# CGRP Induces Differential Regulation of Cytokines from Satellite Glial Cells in Trigeminal Ganglia and Orofacial Nociception

**DOI:** 10.3390/ijms20030711

**Published:** 2019-02-07

**Authors:** Shaista Afroz, Rieko Arakaki, Takuma Iwasa, Masamitsu Oshima, Maki Hosoki, Miho Inoue, Otto Baba, Yoshihiro Okayama, Yoshizo Matsuka

**Affiliations:** 1Department of Stomatognathic Function and Occlusal Reconstruction, Graduate School of Biomedical Sciences, Tokushima University, Tokushima 770-8504, Japan; shaista_afroz@yahoo.com (S.A.); c301551017@tokushima-u.ac.jp (T.I.); m-oshima@tokushima-u.ac.jp (M.O.); hosoki@tokushima-u.ac.jp (M.H.); inoue.miho@tokushima-u.ac.jp (M.I.); 2Department of Oral Molecular Pathology, Graduate School of Biomedical Sciences, Tokushima University, Tokushima 770-8504, Japan; arakaki.r@tokushima-u.ac.jp; 3Department of Oral and Maxillofacial Anatomy, Graduate School of Biomedical Sciences, Tokushima University, Tokushima 770-8504, Japan; baba.otto@tokushima-u.ac.jp; 4Clinical Trial Center for Developmental Therapeutics, Tokushima University Hospital, Tokushima 770-8503, Japan; y-okayama@tokushima-u.ac.jp

**Keywords:** satellite glial cells, calcitonin gene related peptide, cytokine, trigeminal ganglion, thermal hyperalgesia

## Abstract

Neuron-glia interactions contribute to pain initiation and sustainment. Intra-ganglionic (IG) secretion of calcitonin gene-related peptide (CGRP) in the trigeminal ganglion (TG) modulates pain transmission through neuron-glia signaling, contributing to various orofacial pain conditions. The present study aimed to investigate the role of satellite glial cells (SGC) in TG in causing cytokine-related orofacial nociception in response to IG administration of CGRP. For that purpose, CGRP alone (10 μL of 10^−5^ M), Minocycline (5 μL containing 10 μg) followed by CGRP with one hour gap (Min + CGRP) were administered directly inside the TG in independent experiments. Rats were evaluated for thermal hyperalgesia at 6 and 24 h post-injection using an operant orofacial pain assessment device (OPAD) at three temperatures (37, 45 and 10 °C). Quantitative real-time PCR was performed to evaluate the mRNA expression of IL-1β, IL-6, TNF-α, IL-1 receptor antagonist (IL-1RA), sodium channel 1.7 (NaV 1.7, for assessment of neuronal activation) and glial fibrillary acidic protein (GFAP, a marker of glial activation). The cytokines released in culture media from purified glial cells were evaluated using antibody cytokine array. IG CGRP caused heat hyperalgesia between 6–24 h (paired-*t* test, *p* < 0.05). Between 1 to 6 h the mRNA and protein expressions of GFAP was increased in parallel with an increase in the mRNA expression of pro-inflammatory cytokines IL-1β and anti-inflammatory cytokine IL-1RA and NaV1.7 (one-way ANOVA followed by Dunnett’s post hoc test, *p* < 0.05). To investigate whether glial inhibition is useful to prevent nociception symptoms, Minocycline (glial inhibitor) was administered IG 1 h before CGRP injection. Minocycline reversed CGRP-induced thermal nociception, glial activity, and down-regulated IL-1β and IL-6 cytokines significantly at 6 h (*t*-test, *p* < 0.05). Purified glial cells in culture showed an increase in release of 20 cytokines after stimulation with CGRP. Our findings demonstrate that SGCs in the sensory ganglia contribute to the occurrence of pain via cytokine expression and that glial inhibition can effectively control the development of nociception.

## 1. Introduction

Inflammation is a complex biological response, which may be caused by various physiological or pathological conditions affecting different parts of the body. In the peripheral nervous system (PNS), inflammatory pain may depend on the phenomena occurring inside the sensory ganglia. Neurons in sensory ganglia are pseudo-unipolar, as their axons bifurcate and project into the brainstem or spinal cord and the periphery [1], the non-synaptic transmission occurs by released diffusible chemical messengers, such as cytokines [2] and neuron soma is surrounded by a sheath of a distinct type of glial cells called satellite glial cells (SGC). These features, which are unique to the PNS, allow the occurrence of bidirectional communication through the axon to the periphery and center [1], chemical transmission due to neuroinflammatory substances [2], and neuron–glia interaction leading to cross-excitation [3]. Studies involving in vivo and in vitro settings have reported that neurotransmitters such as substance P (SP), calcitonin gene-related peptide (CGRP) or adenosine 5’-triphosphate (ATP)) are released within the sensory ganglia due to inflammatory and neuropathic pain (NP) conditions [4,5]. SGCs respond to sensory ganglion injury or damage by proliferating, expressing the glial fibrillary acidic protein (GFAP) and neuroinflammatory substances and forming gap junctions, thus, sharing the characteristics of glia in the central nervous system [6,7]. In addition, SGCs were reported to have immune properties, as they express mitogen-activated protein kinase (MAPK) and cytokines [8]. In animal models, results have shown that there is an increased activity of the SGCs and an increase in the cytokine level during a pain condition [9,10]. The release of cytokines from the activated glial cells may be responsible for the persistence of pain by causing neuronal excitation. In an in vitro study, exogenous application of IL-1β to neurons of trigeminal ganglion evoked differential responsiveness from neurons. This effect was shown to be mediated by the modification of voltage-gated sodium channels (NaV) and regulated by the MAPK thereby contributing to inflammatory hyperalgesia [11,12]. IL-6 modulates neuronal excitability through NaV1.7, which also involves activation of the MAPK pathway [13]. All these glial factors may help in the regulation of the neuronal microenvironment and the neuronal transmission of pain, thus indicating a link between glial activation and neurogenic stimulation [14]. This entire phenomenon occurs inside the trigeminal ganglion (TG), contributing to neurogenic inflammation and orofacial pain sensation.

Intra-ganglionic (IG) secretion of CGRP modulates the neuronal transmission of pain signals [15,16,17,18]. CGRP is a potent neuroinflammatory mediator, contributing to the development of peripheral and central sensitization in orofacial inflammatory, neuropathic pain, migraine and medication overuse-related headache [15,16,19] and a key mediator of neuroimmune communication [17]. In TG, CGRP is primarily found in the small-medium-diameter neurons whereas CGRP receptors are found on large diameter neurons and SGCs [18,20]. Moreover, CGRP can induce its own expression in the trigeminal neurons via a protein kinase A-mediated pathway [21]. Thus, the CGRP released within the TG can locally affect SGCs and sensitize the primary afferent neurons, which in turn can activate positive feed-forward circuitries that can initiate and or sustain a painful event. Therefore, blocking this loop may have a therapeutic effect [22].

The aim of the present study was to investigate the role of the satellite glial cells in TG on cytokine-related nociception in response to IG administration of CGRP.

## 2. Results

### 2.1. Effect of Intra-Ganglionic Calcitonin Gene-Related Peptide (CGRP)

#### 2.1.1. Intra-Ganglionic CGRP Mediates Orofacial Thermal Hyperalgesia

The reward-licking events/face-contact events ratio (L/F) and stimulus duration/face-contact events were evaluated after IG drug administration and compared with the baseline. These parameters depend on the contact made by an animal with the thermode to reach the reward bottle. As the temperature becomes aversive or less tolerable, the animal withdraws more frequently, thus, decreasing the L/F ratio and the stimulus duration/face-contact events (seconds). Across the examined temperature range, both the L/F ratio and the stimulus duration/face-contact events (seconds) were significantly reduced at 45 °C 6 h post-CGRP injection and only L/F ratio 24 h post-CGRP injection, compared to baseline. This result indicates that tolerance to heat decreases 6 and 24 h after IG CGRP administration (Figure 1). Six hours after CGRP administration the L/F ratio and stimulus duration/face-contact events (seconds) were nearly the same at 37 °C. After 24 h, there was an increase in both the values but this effect could not reach statistical significance. The effect of CGRP administration on allodynia was evaluated by assessing both behavioral outcomes at 10 °C. No significant behavioral changes were observed after CGRP administration compared to the baseline at the same temperature after 6 and 24 h.

#### 2.1.2. Intra-Ganglionic CGRP-Induced Thermal Hyperalgesia Is Accompanied by Satellite Glial Cell Activation in Trigeminal Ganglion (TG)

Several studies have reported that GFAP, an intermediate filament in the cytoplasm, is a marker of glial cell activation [23,24,25]. Although in normal resting conditions, SGCs do not express GFAP, they do so in response to any kind of injury. In the present experiment, glial activation showed a time-related change in both mRNA and protein expression after CGRP administration. Between 1 and 6 h, GFAP mRNA expression was significantly higher in the CGRP-injected group than in the control group, (Figure 2a). The mRNA expression of GFAP decreased 24 h after CGRP administration, although it did not reach the basal level. This change in mRNA expression was concomitant with an increase in GFAP protein expression, occurring 1-6 h after CGRP injection, (Figure 2b,c). Both indicate an increase in glial activity, occurring concomitantly to thermal hyperalgesia at 45 °C, 6 h post-administration.

#### 2.1.3. Intra-Ganglionic CGRP-Induced Thermal Hyperalgesia Is Accompanied by Differential Regulation of Cytokines in TG

Circulating cytokines are known to be involved in the inflammatory pain phenomenon, and indirect evidence suggests that the cytokines produced inside the ganglion are also involved in pain initiation and sustainment [23]. To investigate the CGRP-induced cytokine modulation inside the TG, we evaluated the mRNA expression of three pro-inflammatory and one anti-inflammatory cytokine, IL-1β, IL-6, and TNF-α and IL-1RA, in the TG tissues after IG CGRP administration. The expression of the pro-inflammatory cytokines IL-1β and IL-6 increased between 1 and 6 h compared to the control group (Figure 3a). However, statistical significance was reached only in the expression level of IL-1β 6 h after CGRP injection. This significant increase in IL-1β expression coincided with the thermal hyperalgesia, occurring 6 h after CGRP injection and glial activation. However, 24 h later the expression level of IL-1β and IL-6 were nearly similar to the control group. The mRNA expression of TNF-α, a very potent pro-inflammatory cytokine, did not differ significantly between the treatment and control groups, from 1 to 24 h (Figure 3a). The mRNA expression of anti-inflammatory cytokine IL-1RA was upregulated between 1–6 h, reaching statistical significance 6 h after CGRP injection. This coincided with the peak glial activity. A comparison between ipsilateral and contralateral sides showed increased expression on the injected side at 1 and 6 h for IL-1β, IL-6, and IL-1RA. However, a statistically significant result was observed only for IL-6 at 6 h (Figure 3b).

#### 2.1.4. CGRP Induces Differential Regulation of Various Cytokines from the Glial Cells

To validate the findings of the differential mRNA expression of cytokines in TG and to confirm their source as glial cells, an in vitro experiment was performed. Overnight incubation of glial rich culture with CGRP induced the release of various cytokines, as tested using an antibody cytokine array (Figure 4). Overnight stimulation (12 h) of glial rich culture with 1 µM CGRP led to a more than 1.5-fold increase in the protein expression of 20 cytokines, no change in 6 cytokines (between 1–1.5-fold change) and down-regulation of 3 cytokines, as compared to control conditions (glial rich culture exposed to only serum-free medium) (Table 1). Up-regulated cytokines included IL-1β, IL-6, and IL-1RA. The expression level of TNF-α remained unchanged after CGRP stimulation.

### 2.2. Effect of Injecting Minocycline (Min) Intra-Ganglionic 1 Hour before Injecting CGRP

#### 2.2.1. Min Prevented the Pro-Nociceptive Effect of Intra-Ganglionic CGRP

In the present experiment, the effect of Minocycline (Min) IG administration 1 h prior to CGRP on the sensitivity to neutral, hot and cold temperatures was investigated. Within the first 6 h, Min administration 1 h before CGRP increased the L/F ratio and stimulus duration/face-contact events (seconds) for all temperatures (i.e., 37, 45 and 10 °C), compared to both baseline and administration of CGRP alone, (Figure 5). This increase was statistically significant at 45 °C around 6 h, however at 24 h it was not statistically significant There was nearly similar behavior outcome for the group receiving Min + CGRP at 37 °C after 24 h compared to the baseline. After 24 h at 10 °C, a decrease in L/F ratio and stimulus duration/face-contact events (seconds) were reduced after Min injection, but it was not statistically significant.

#### 2.2.2. Min Prevented the Pro-Nociceptive Effect of Intra-Ganglionic CGRP via Inhibition of Glial Activation

Effect of Min on the glial inhibition was investigated by intra-ganglionic administration of Min 1 h before CGRP and evaluation of mRNA and protein expression of GFAP. The Min + CGRP injected group exhibited a significantly decreased GFAP mRNA expression between 1 and 6 h, compared to the group administered CGRP alone, Figure 6a. This decrease occurred concomitantly with a reduction in GFAP protein expression Figure 6b,c. After 24 h, both groups presented a nearly similar GFAP expression, which could not reach the basal level. The decrease in glial activity between 1 and 6 h, as noted by the decreased GFAP mRNA and protein expression, coincided with increased L/F ratio and stimulus duration/stimulus-contact events (seconds) at 45 °C at 6 h in Min + CGRP injected group, thus, indicating that glial inhibition contributes to increased tolerance to heat.

#### 2.2.3. Min Induced Glial Inactivation Is Accompanied by Reduced Expression of Cytokines in TG

Between 1 and 6 h, Min reduced the CGRP-induced expression of cytokines IL-1β, IL-6, and IL-1RA, which was statistically significant for IL-1β and IL-6 around 6 h after injection (Figure 7). No significant difference was observed between the CGRP and Min + CGRP injected groups regarding the expression of any cytokine after 24 h. There was no difference in the expression of TNF-α in either group.

### 2.3. CGRP-Induced Neuronal Activation Is Prevented by Min

Studies involving in vivo and in vitro setting have shown that glial secreted cytokines cause neuronal excitation. This neuronal excitability may be due to the modification of NaV channels, out of which NaV1.7 has been reported to be involved [11,12,13]. In the present experiment, neuronal excitation was examined by a change in the expression of NaV1.7 after CGRP and Min + CGRP administration. In the sensory ganglion, NaV1.7 is primarily distributed primarily in small-diameter neurons, which also express the CGRP and TRPV1 receptors—a thermal sensor and its activation is responsible for the detection of painful thermal stimuli [26,27]. Exogenously administered CGRP significantly increased the expression of NaV1.7 at 6 h, whereas Min significantly reduced the expression of NaV1.7 at 6 h, (Figure 8). Thus, in the present study, a temporal concordance was noted between glial activation and neuronal activation, apart from that glial inhibition resulted in reduced neuronal activity.

## 3. Discussion

The neuron–glia cross talk and the role of satellite glial cells in the maintenance of pain are a well-known concept in pain research. The evidence provided by the present study supports this concept, by showing that the IG administration of CGRP in the TG caused heat hyperalgesia, extending between 6 and 24 h post-administration. During 24 h behavior assessment at 37 °C, both L/F ratio and stimulus duration/face-contact showed an increase in the trend beyond the baseline. It cannot be confirmed based on the results that this effect was due to analgesia or reduced stress to the animals. However, this effect did not reach statistical significance. Increased glial activity was observed during the first 6 h, as evaluated by increased mRNA and protein expression of GFAP, and the increased mRNA expression of the pro-inflammatory cytokine IL-1β and the anti-inflammatory cytokine IL-1RA in the TG tissues. In our cell culture experiment, CGRP stimulation resulted in an increased expression of 20 cytokines, as detected by the antibody cytokine array, including IL-1β, IL-6, and IL-1RA. When Min, a glial inhibitor drug, was injected one hour before CGRP administration, L/F and stimulus duration/face-contact increased significantly at 45 °C after 6 h compared to the CGRP injected group. A significant reduction of glial activity was observed between 1 and 6 h in the Min + CGRP injected group, as detected by the down-regulation of the GFAP expression and a significant fall in mRNA expression of cytokines IL-1β and IL-6 compared to the only CGRP injected group. However, at 24 h, cold sensitivity showed a reduction, as measured by the L/F and average contact duration in the Min + CGRP group, although this effect was not statistically significant. In the present study, no difference was observed among the CGRP, Min + CGRP and saline groups, regarding TNF-α mRNA expression. Additionally, the cell culture experiment showed that overnight stimulation with CGRP did not change the expression of TNF-α compared to the control condition. In the CGRP-injected group, NaV1.7 was upregulated 6 h post-administration, whereas in the Min + CGRP-injected group it was down-regulated.

A previous animal study showed that the incubation of isolated SGCs with fractalkine induced the release of TNF-α, IL-1β, and PGE-2 [23]. Differential cytokine regulation was detected with an antibody array when the glial rich culture was stimulated with CGRP [8,9,28]. A previous study using rat TG primary cultures reported that 1 µM CGRP was associated with a modest pro-inflammatory effect by upregulating the expression of inflammatory genes at the mRNA level, without modifying the secretion of proinflammatory mediators, including IL-1β [29]. However, CGRP potentiated the level of glial activation induced by IL-1β, thus, showing that SGCs are activated by CGRP released from neurons, and CGRP + cytokines can have a synergistic effect on glial activation, thus forming a positive feedback loop within the TG [29,30]. Another study demonstrated the co-expression of IL-1β/GFAP immunoreactivity in TG satellite cells and a paracrine mechanism of action of IL-1β, inducing neuronal hyperexcitability in response to inflammation [2,31]. Stimulation of human TG suspension of SGC with specific toll-like receptor 1-5 ligand caused IL-6 and TNF-α release in the supernatant [7].

From the results of the present study, it cannot be determined whether the presence of thermal hyperalgesia at 24 h was due to a peripheral effect or to central sensitization. It has been reported previously that injecting CGRP or olcegepant (CGRP receptor antagonist) in the TG does not affect the activity of spinal trigeminal neurons [32]. However, this effect was examined within 20 min of each treatment. When released at extravascular sites, CGRP can have an extremely long duration of action [33]. In migraine patients, intravenous infusion of human α-CGRP requires a considerable amount of time to provoke migraine-like headaches [22,34]. In addition, our findings indicate that the presence of thermal hyperalgesia between 6 and 24 h post-treatment may involve an unidentified intracellular mechanism, an extended receptor activation, the endogenous release of CGRP or the sensitization of TG afferents and/central site that receive input from TG [21,22,33,34].

Intrathecal administration of Min has been found to inhibit low-threshold mechanical allodynia by decreasing microglial activation and simultaneously decreasing the mRNA expression of IL-1β, TNF-α, IL-1 RA and IL-10 in the lumbar and dorsal spinal cord, and the expression of IL-1β and TNF-α in the cerebrospinal fluid [35]. Intrathecal co-injection of Min and CCL-2 has been found to block the CCL-2 induced reduction in the latency of hind-paw withdrawal and thermal hyperalgesia [36]. In a model of sciatic nerve ligation neuropathic pain, Min induced reversal of hyperalgesia and allodynia by inhibiting IL-6 production [37]. These findings are consistent with our results.

In our experimental condition, TNF-α mRNA and protein expression were relatively unchanged in all the groups. In antibody cytokine array analysis studies, CGRP stimulation of glial rich culture showed different results in the fold change of TNF-α by different researchers: >3-fold increase in TNF-α [8], 50% of control [9], no significant change [28]. This difference in result may be due to the difference in animal species, the age of the animal used for the experiment, cell density, the concentration of CGRP or time of stimulation etc. According to a previous study, CGRP-mediated immunosuppressive activity is due to suppression of TLR-stimulated dendritic cells TNF-α production by a mechanism involving rapid up-regulation of the transcriptional repressor inducible cAMP early repressor [38]. In contrast, in an organ culture study of TG, TNF-α mRNA showed highly significant upregulation when co-incubated with CGRP, which was counteracted by the addition of CGRP_8-37_ (CGRP antagonist) [39]. TNF-α induces pro-inflammatory signal cascades accompanied with an increase in the synthesis and release of CGRP by trigeminal ganglion neurons [40] and in migraineurs, serum TNF- α and IL-1β are shown to be elevated during the attack [41].

There is evidence indicating that glial-secreted IL-1β upregulates the expression of neuronal NaV 1.7 in the sensory ganglia, thus, contributing for inflammatory hyper-nociception, whereas glial inhibitors block the inflammation-induced SGCs activation, thus, alleviating inflammation-induced hyper-nociception [42]. This is consistent with our results since we observed that NaV1.7 upregulation was accompanied by glial activation, IL-1β upregulation, and thermal hyperalgesia. Moreover, all these effects were reversed by the injection of Min 1 h prior to CGRP administration. The results of this study indicate the occurrence of neuronal activation in response to glial-activated secretion of pro-inflammatory substances and neuron-glia cross-excitation after IG CGRP injection. In animal studies with knockout/knockdown of NaV1.7, a reduced response to inflammatory hyperalgesia has been observed [43,44]. In humans, mutations in SCN9A gene (encoding NaV1.7) are related to pain disorders [45,46].

CGRP is a neuropeptide that plays diverse actions in different parts of the body and several studies have investigated previously the effect of exogenous CGRP on behavior. In a study with healthy volunteers, intravenous infusion of CGRP induced mild headache, whereas stimulation of V1 trigeminal area with noxious heat caused increased neuronal activity in the brainstem and insula, decreasing neuronal activity in the caudate nuclei, thalamus and cingulate cortex [47]. Microinjection of CGRP into different areas of the nervous system resulted in differential behavior responsiveness ranging from increased to decreased nociception [48,49,50,51,52,53,54].

This study has certain limitations. Apart from glial inhibition, Min has a neuroprotective activity by reducing the expression of metalloproteinases, and anti-inflammatory activity by inhibiting circulating macrophages [55,56]. This study did not probe into these possibilities. The therapeutic effect of Min was not tested by administering Min after CGRP-induced inflammation. Peltier rods of the orofacial pain assessment device (OPAD) were programmed to have a similar bilateral temperature cycle. Therefore, a conclusion about the differential effect of the drug in causing hyperalgesia to the ipsilateral and contralateral side cannot be drawn. However, according to one study, unilateral injection of CGRP in hind paw had a bilateral effect due to local and neurogenic inflammatory mechanism, and endogenous secretion of CGRP [57]. 

To the best of the authors’ knowledge, this is the first study showing the effect of exogenous IG CGRP in TG on orofacial thermal nociception and the interplay of cytokines and neuron–glia interaction. Taking all the findings together, we can conclude that SGCs are involved in pain modulation by augmenting and sustaining inflammatory processes in the sensory ganglia and that glial inhibition can be used to effectively treat neurogenic inflammation-associated pain (Figure 9).

## 4. Materials and Methods

### 4.1. Animals

The Animal Research Committee of the Tokushima University approved all animal care and experimental procedures (Protocol number- T27-78, date 4 November 2015; T30-75 date 27 September 2018). For in vivo experiments, 4-week old male Sprague Dawley rats were habituated to the animal housing room for 2 weeks and were used for drug administration and behavioral assessment between 6 and 9 weeks of age, at the body weight of 230–280 g. The rats were housed in groups of 2 or 3 rats per cage, under a controlled light cycle (lights on at 6:00 and off at 18:00). Food and water were available ad libitum, except when the animals fasted for behavior testing. All efforts were made to minimize animal suffering and to reduce the number of animals used. All the experiments were performed between 9:00 and 16:00.

### 4.2. Anesthesia

Medetomidine (Nippon Zenyaku Kogyo, Fukushima, Japan) 0.375 mg/kg, midazolam (Sandoz K.K., Yamagata, Japan) 2.0 mg/kg and butorphanol (Meiji Seika Pharma Co., Ltd., Tokyo, Japan) 2.5 mg/kg were administered intraperitoneally to induce anesthesia according to a previously published protocol [58]. Rapid recovery from anesthesia was achieved by administration of atipamezole (0.75 mg/kg), an antagonist of medetomidine (Nippon Zenyaku Kogyo Co., Ltd., Tokyo, Japan). Prior to euthanasia, the animals were deeply anesthetized using sodium pentobarbital (Kyoritsu Seiyaku Corporation, Tokyo, Japan) 75 mg/kg, administered intraperitoneally.

### 4.3. Intra-Ganglionic Drug Administration

The rat α-CGRP (R&D System, Minnesota, MN, USA) solution and Min (Nichi-Iko, Toyama, Japan) were prepared in normal saline, aliquoted and stored at −20 °C, following the manufacturer’s instructions. For TG IG drug administration, either CGRP alone (10 μL of 10^−5^ M) [32], Min (5 μL containing 10 μg Min) followed by CGRP with one hour gap (Min + CGRP), or normal saline (10 μL) were injected according to a previously published protocol [59]. Briefly, IG administration was performed using a 26 gauge, 10 μL Hamilton syringe. The canal was accessed from the infra-orbital foramen, 1 mm medial to the zygomatic process of the maxilla, and the needle was inserted approximately 22 mm, at an angle of approximately 10° toward the midline and approximately 15° downward from the plane formed by the parietal bone [59]. The needle traversed through the infraorbital canal into the ipsilateral trigeminal ganglion located in the Meckel’s cave. Direct blue 1 (TCI, Tokyo, Japan) was injected to confirm the correct location of the injection in the TG.

### 4.4. Behavioral Assessment

After IG drug administration, rats were evaluated for thermal hyperalgesia at both 6 and 24 h post-injection [23]. The evaluation was performed in independent experiments, using an operant orofacial pain assessment device (OPAD, Stoelting Co., Wood Dale, IL, USA, which was a kind gift from Dr. John Neubert, University of, Gainesville, FL, USA). All the rats were tested only once after drug administration. The rats were tested on the OPAD at three temperatures (i.e., 37, 45 and 10 °C), within a 15 min behavioral session. A ramping cycle of 60 s, remaining at a temperature for 60 s, was used during the entire 15 min session, which allowed for testing at the neutral temperature and for assessing pain at hot and cold temperatures within a single testing session [60]. The rats were trained at a neutral temperature (i.e., 37 °C) to press their faces against temperature-controlled Peltier rods to gain access to a food bottle filled with diluted sweetened condensed milk (1:2, milk to water) (Morinaga, Tokyo, Japan) as a reward until they licked at least 1000 times during a 10-min session. The rats were shaved 1–2 days prior to testing and fasted 18 h before the baseline recording (on 3 different days with a gap of at least 2 days between each session) and after IG injections (one time). 

### 4.5. Quantitative Real-Time Polymerase Chain Reaction (qRT-PCR)

TGs were dissected after IG drug administration, at 1, 6 and 24 h post-injection and were immediately stored in RNA later (Sigma-Aldrich, St. Louis, MO, USA) following the manufacturer’s instructions. Saline-injected rats were used as a control. A tissue lyser was used to homogenize the tissues. Total RNA was extracted from the trigeminal ganglion using trizol reagent (Invitrogen, Carlsbad, CA, USA) and reverse-transcribed to cDNA using a high-capacity cDNA reverse transcription kit (Applied Biosystems, Foster City, CA, USA) following the manufacturer’s instructions. Quantitative real-time polymerase chain reaction (qRT-PCR) for seven genes was performed to evaluate the expression of Tbp (TATA box binding protein), IL-1β, IL-6, TNF-α, IL-1RA, NaV 1.7 and GFAP mRNA. The following primer sequences were selected from previously published research (F, forward; R, reverse):

IL-1β F_5′-cacctctcaagcagagcacag-3′, IL-1β R_5′-gggttccatggtgaagtcaac-3′ [61], IL-6 F_5′-tcctaccccaacttccaatgctc-3′, IL-6 R_5′-ttggatggtcttggtccttagcc-3′ [61], TNF-α F_5′-aaatgggctccctctcatcagttc-3′, TNF-α R_5′-tctgcttggtggtttgctacgac-3′ [61], Na_V_1.7 F_5′- tcgtaccccatagaccccg-3′, Na_V_1.7 R_5′-ctgattagtcgtgccgctg-3′ [62], IL-1RA F_5′-gagacaggccctaccaccag-3′, IL-1RA R_5′-cgggatgatcagcctctagtgt-3′ [63], GFAP F_5′-agtggtatcggtccaagtttgc-3′, GFAP R_5′-tggcggcgatagtcattagc-3′ [64], Tbp F_5′-tgggattgtaccacagctcca-3′, Tbp R_5′-ctcatgatgactgcagcaaacc-3′ [65].

Real-time PCR was performed using TB Green premix Ex Taq II (Takara Bio Inc., Kusatsu, Shiga Prefecture, Japan) in a DNA thermal cycler (ABI 7300, Applied Biosystems, Foster City, CA, USA), according to the following protocol: denaturation step- 95 °C for 30 s; 40 cycles of 95 °C for 15 s, annealing for 20 s, and annealing temperature at 60 °C for 30 s. Dissociation curve analysis was performed to confirm the specificity of the final product. Gene expression quantification was based on the *C*_T_ values. The expression levels were normalized to an endogenous control, Tbp expression, and the ratios were multiplied by 100.

### 4.6. Immunohistochemistry

Six to 8 microns-thick frozen sections of trigeminal ganglion were fixed in 4% paraformaldehyde, permeabilized with 0.1% Triton and subsequently blocked with Blocking One Histo (Nacalai Tesque, Kyoto, Japan). Sections were incubated with primary antibodies, rabbit anti-glutamine synthetase (1:1000; ab49873, Abcam, Cambridge, UK) and goat anti-GFAP (1:500; ab53554, Abcam), overnight at 4 °C. Sections were incubated with secondary antibodies for 2 h at room temperature using donkey anti-goat IgG Alexa Fluor 488 (1:200; ab150129, Abcam) and donkey anti-rabbit Alexa Fluor 555 (1:200; ab 150074, Abcam). For nuclear staining, 4’, 6-Diamidino-2-phenylindole dihydrochloride (DAPI, 1:100, Nacalai Tesque, Inc., Kyoto, Japan) was used for 15 min at room temperature, followed by mounting with Aqua-Poly/Mount (Polysciences, Inc., Warrington, PA, USA). Isotypes (goat IgG bs-0294P, and rabbit IgG bs-0295P, Bioss Antibodies, Boston, MA, USA) and only secondary antibodies were used as positive and negative control. Images were observed and acquired by confocal laser-scanning microscope (LSM 700, Carl Zeiss, Oberkochen, Germany).

### 4.7. Cytokine Measurement

The TGs were dissociated as described previously [66]. Briefly, the tissues were enzymatically digested using 0.125% collagenase P (Roche, Indianapolis, IN, USA), 0.02% DNase (Sigma-Aldrich, St. Louis, MO, USA) and 0.25% trypsin (Sigma, Kanagawa, Japan), and then mechanically triturated with fire-polished Pasteur pipettes in dissociation solution (5 mL Hank Balanced Salt Solution (Nacalai Tesque, Inc, Kyoto, Japan) containing 0.295% MgSO_4_ and 0.02% DNase). Dissociated cells were plated on uncoated 24-well plates in feeding medium containing minimum essential medium (MEM) (Life Technologies, Tokyo, Japan), fetal bovine serum (Sigma-Aldrich, St. Louis, MO, USA), glucose, transferrin, glutamine, and insulin (Sigma-Aldrich, St. Louis, MO, USA). The primary mixed neuron-glial cell culture was maintained for 5 days. The glial rich culture was prepared from the mixed culture at day 5, by detaching the cells by 5-min treatment with acutase (Nacalai Tesque, Inc., Kyoto, Japan) at 37 °C and re-plating on an uncoated 24-well plate for 48 h (Figure 10) [28].

The cytokine released in culture media from purified SGC cultures was evaluated under control conditions (i.e., samples exposed to MEM alone) and following overnight exposure (12 h) to 1 μM CGRP [28]. Cytokine measurement of conditioned medium derived from glial rich culture exposed to CGRP and control conditions was performed using the Proteome profiler Rat Cytokine Array Panel A (R&D System, Minneapolis, MN, USA) according to the manufacturer’s instructions. This array kit allows to simultaneously detect 29 different rat cytokines and chemokines. Spot densitometry was performed using Chemi Doc systems (BioRad, Hercules, CA, USA). Background staining and spot size were analyzed and the results were expressed as the fold change of CGRP-treated cultures in relation to control conditions.

### 4.8. Statistical Analysis

SPSS 25 (IBM, New York, NY, USA) was used to perform statistical analysis. All the results were expressed as mean ± standard error of mean (SEM). Within-group differences were examined using the “paired *t*-test”. mRNA expression was compared using one-way analysis of variance (ANOVA), followed by Dunnett’s post hoc test. The “*t* test” was used to examine between-group differences. A level of *p* < 0.05 was considered statistically significant.

## 5. Conclusions

In conclusion, we found that IG CGRP injection induces increased sensitivity to heat. This effect was accompanied by an increased SGC and neuronal activation. An increased expression of cytokine during the same time within the TG affirm their contribution in the genesis of pain. Additionally, injecting the glial inhibitor Minocycline 1 h before CGRP decreased the CGRP induced thermal hyperalgesia, SGC activation and reduced the expression of pro-inflammatory cytokines IL-1β and IL-6 in the TG. Taken together, these findings support the notion that increased glial activity contributes to hyperalgesia and that glial inhibition can be considered for its management.

## Figures and Tables

**Figure 1 ijms-20-00711-f001:**
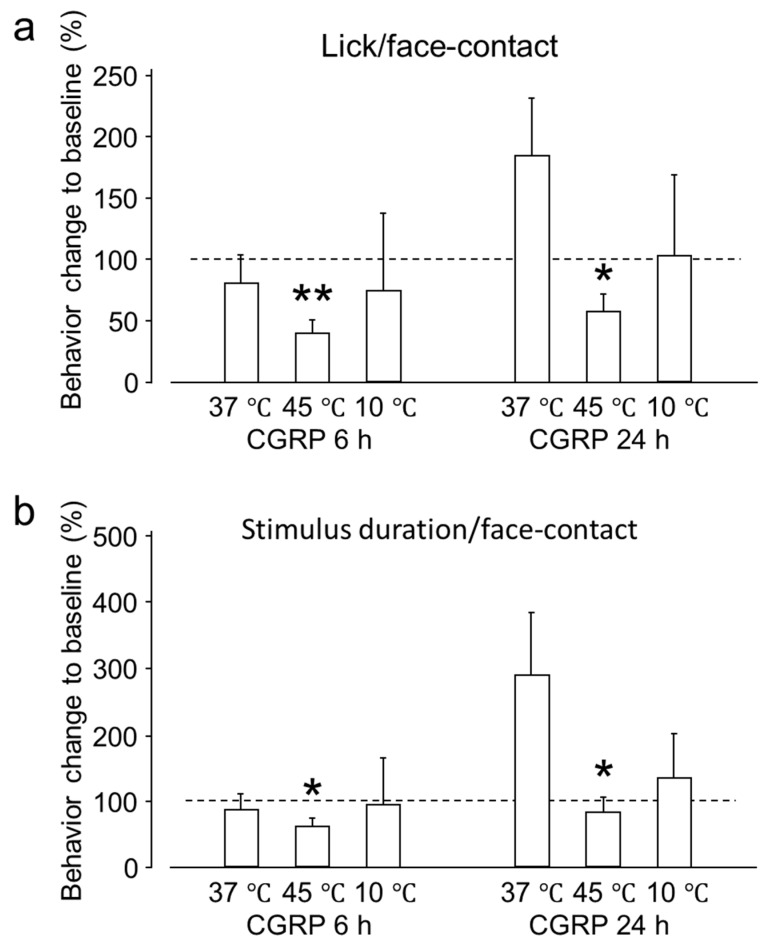
Behavioral outcome of intra-ganglionic (IG) calcitonin gene-related peptide (CGRP) administration. (**a**) The reward-licking events/face-contact events (L/F) ratio (mean ± standard error of the mean (SEM)) was significantly reduced at 45 °C, both at 6 and 24 h after IG CGRP administration. (**b**) The stimulus duration/face-contact (s) (Mean ± SEM) was significantly reduced at 45 °C, at 6 h after IG CGRP administration. These results indicate that CGRP decreases the tolerance to heat between 6 and 24 h. Results are presented as percent change from untreated baseline at the respective temperature. *: *p* < 0.05, **: *p* < 0.01 with paired-*t* test. *n* = 7 rats were assigned to each group.

**Figure 2 ijms-20-00711-f002:**
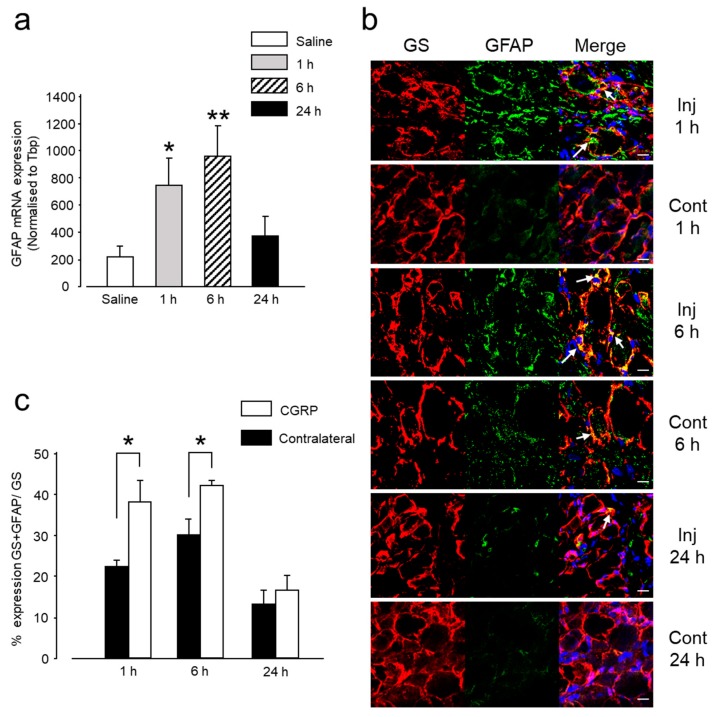
IG CGRP induced satellite glial cells (SGCs) activation. (**a**) The mRNA expression of glial fibrillary acidic protein (GFAP) in the trigeminal ganglion (TG) was significantly increased at 1 and 6 h after IG CGRP administration. Results are presented as Mean ± SEM of the relative expression. *: *p* < 0.05, **: *p* < 0.01 with one-way analysis of variance (ANOVA) followed by the Dunnett *t* test. *n* = 5 rats were assigned to each group. (**b**) Confocal images of immunofluorescent staining of TG sections with glutamine synthetase (GS, red), GFAP (green), and 4’,6-Diamidino-2-phenylindole dihydrochloride (DAPI, blue) at 1, 6 and 24 h after IG CGRP administration and contralateral TGs. Colocalization of GS and GFAP in the SGCs is denoted by white arrow. Scale bar: 20 µm. (**c**) IG CGRP administration increased the GFAP protein expression on the injected side compared to the contralateral side both at 1 and 6 h. *: *p* < 0.05, with *t*-test. *n* = 3 rats were assigned to each group and data were acquired from three independent sections (i.e., examined in three non-overlapping views).

**Figure 3 ijms-20-00711-f003:**
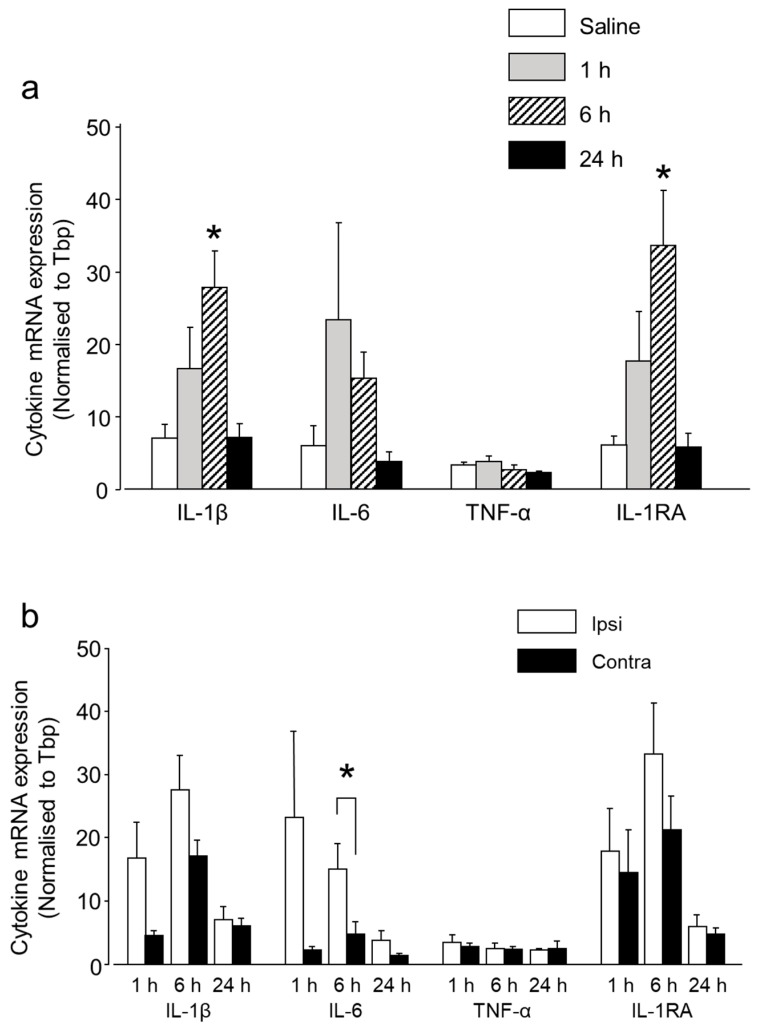
Differential mRNA expression of cytokines after IG CGRP administration. (**a**) IG CGRP increased the expression of IL-1β, IL-6 and IL-1RA at 1 and 6 h after administration compared to the saline-injected group. However, the statistically significant result was observed only for IL-1β and IL-1RA after 6 h. After 24 h, the expression of IL-1β, IL-6 and IL-1RA decreased. TNF-α expression was similar in all the groups. #: *p* < 0.05 with one-way ANOVA followed by the Dunnett *t* test for comparison with saline injection. *n* = 5 rats were assigned to each group. (**b**) A comparison between ipsilateral and contralateral side showed increased expression on the injected side at 1 and 6 h for IL-1β, IL-6 and IL-1RA. However, statistically significant result was observed only for IL-6 at 6 h using one-way ANOVA, *: *p* < 0.05, *n* = 5 in each group.

**Figure 4 ijms-20-00711-f004:**
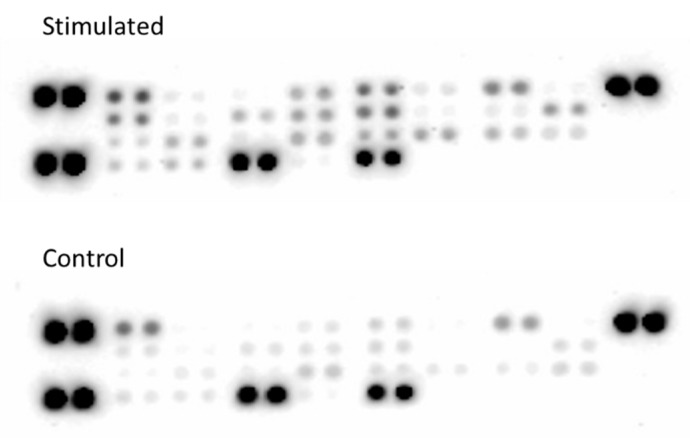
Cytokine array membrane. Protein profiling in cell culture supernate in stimulated and control condition. Overnight stimulation of glial-rich culture with 1 μM CGRP causes differential expression of cytokines compared to control condition.

**Figure 5 ijms-20-00711-f005:**
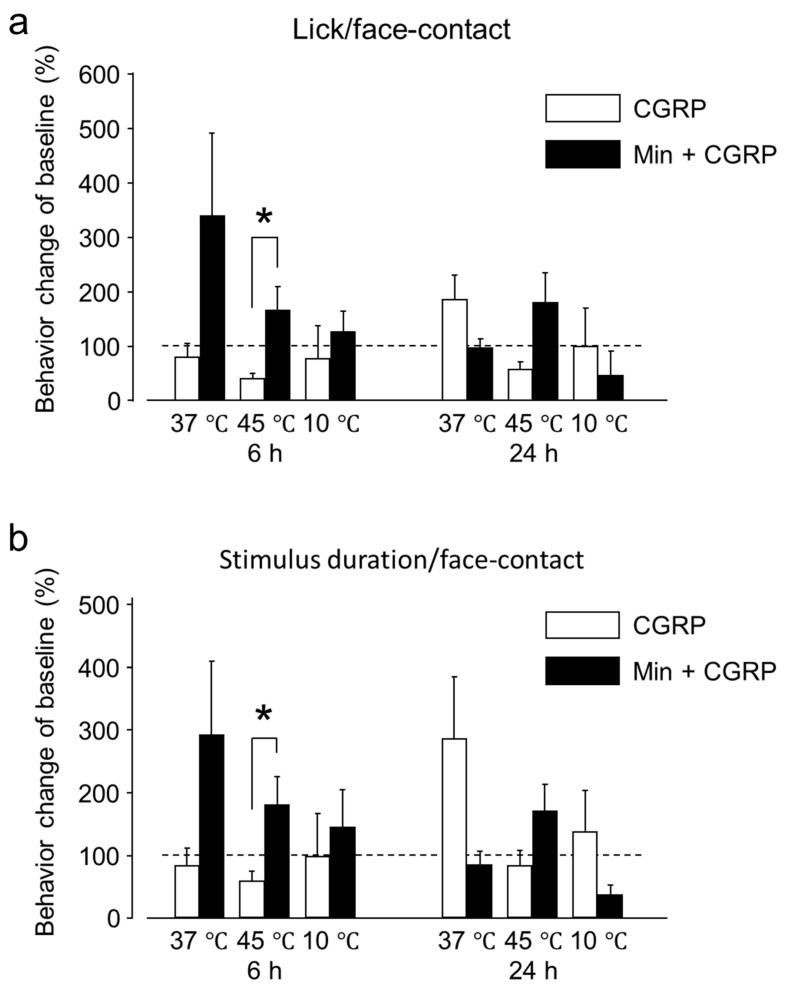
Behavioral outcome to IG CGRP and Minocycline (Min) + CGRP administration. (**a**) The L/F ratio (Mean ± SEM) was significantly increased at 45 °C, 6 h after administration in the Min + CGRP injected group. (**b**) The stimulus duration/face-contact (s) (Mean ± SEM) were significantly increased at 45 °C, 6 h after administration in the Min + CGRP injected group. Results are presented as percent change from untreated baseline at the respective temperature. *: *p* < 0.05 with *t*-test. *n* = 7 rats were assigned to each group.

**Figure 6 ijms-20-00711-f006:**
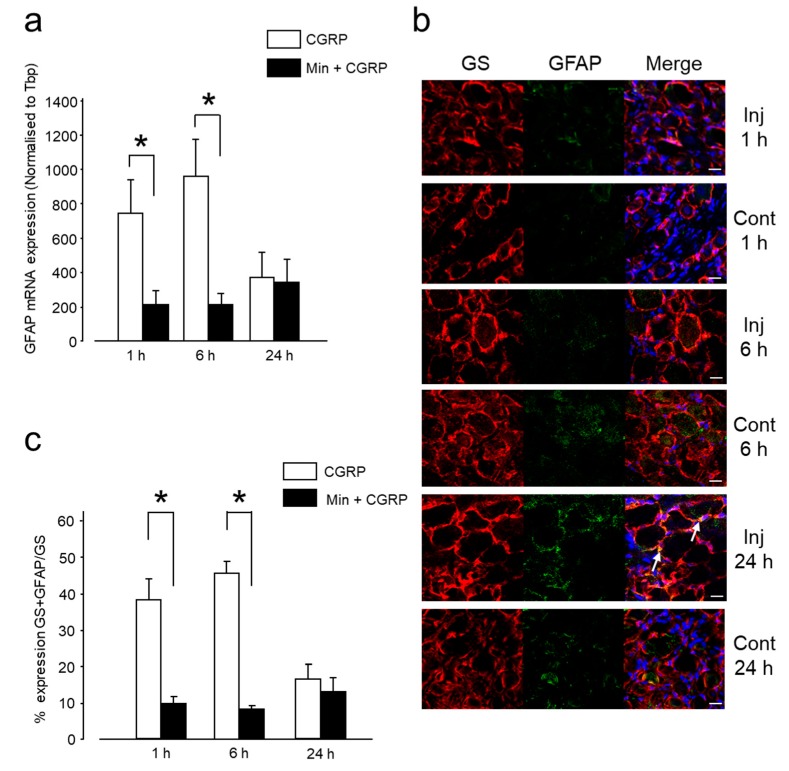
Effect of injecting Min 1 h before CGRP on glial inhibition. (**a**) mRNA expression of GFAP in TG was significantly decreased 1 and 6 h after administration in the Min + CGRP group compared to CGRP alone group. Results are presented as Mean ± SEM of relative expression. *: *p* < 0.05 with *t*-test. *n* = 5 rats were assigned to each group. (**b**) Confocal images of immunofluorescent staining of TG sections with GS (red), GFAP (green), and DAPI (blue) after 1, 6 and 24 h of IG Min + CGRP administration and contralateral TG. Colocalization of GS and GFAP in the SGCs is denoted by white arrow. Scale bar: 20 µm. (**c**) IG Min 1 h before CGRP administration decreased the GFAP protein expression compared to only CGRP injection at 1 and 6 h. *: *p* < 0.05 with *t* test. *n* = 3 rats were assigned to each group and data were acquired from three independent sections (i.e., examined in three non-overlapping views).

**Figure 7 ijms-20-00711-f007:**
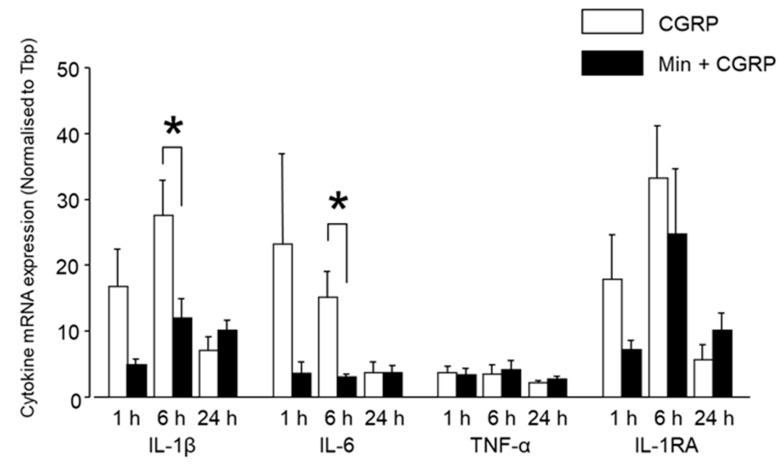
Differential mRNA expression of cytokines after IG CGRP and Min + CGRP administration. IG Min administered 1 h before CGRP decreased the expression of IL-1β, IL-6 and IL-1RA after 1 and 6 h, compared to CGRP alone. After 24 h, there was a decrease in the expression of IL-1β, IL-6 and IL-1RA in bothe the groups. Min administration had no effect on TNF-α expression. *: *p* < 0.05 with *t*-test. *n* = 5 rats were assigned to each group.

**Figure 8 ijms-20-00711-f008:**
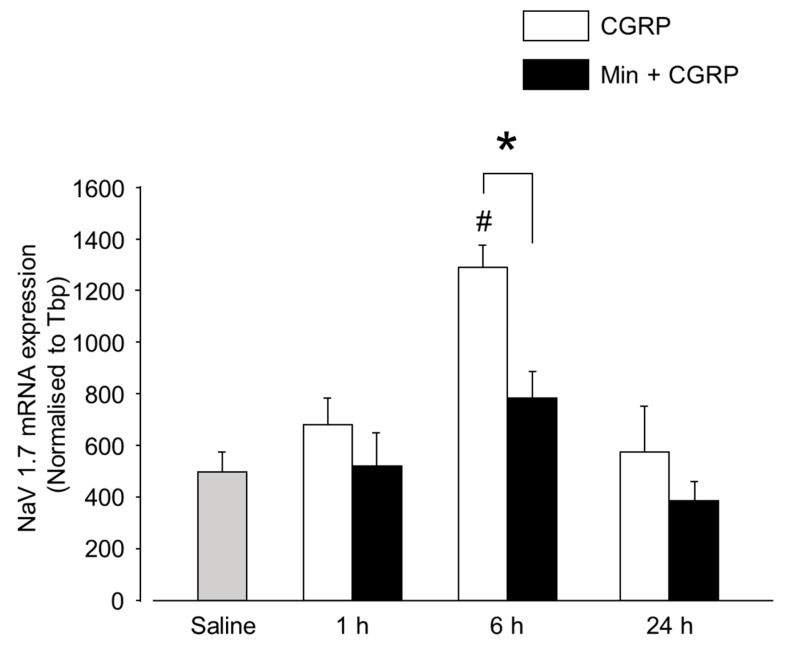
Effect of IG CGRP and Min + CGRP on NaV1.7 expression. CGRP increased the mRNA expression of NaV1.7 after 6 h and Min reduced the CGRP induced upregulation of NaV1.7 expression. *: *p* < 0.05 with *t* test and #: *p* < 0.05 with one-way ANOVA followed by the Dunnett *t* test for comparison with saline injection. *n* = 5 rats were assigned to each group.

**Figure 9 ijms-20-00711-f009:**
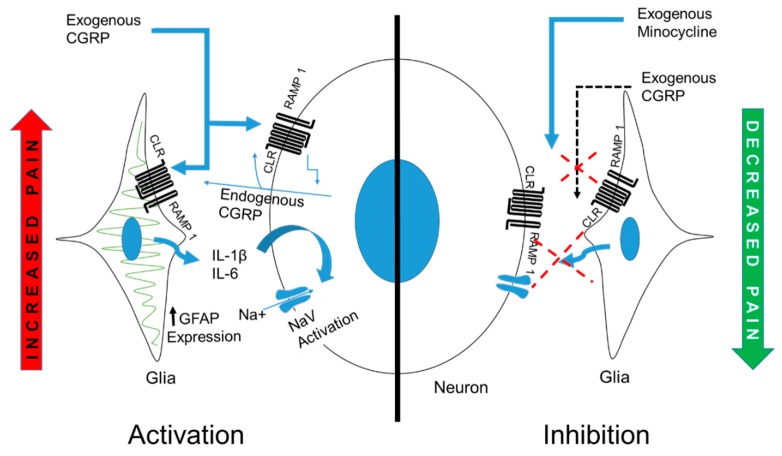
Schematic representation of the effect of exogenously administered CGRP on neurons and SGCs. CGRP receptors are present on the SGCs and neurons [namely, receptor activity-modifying protein 1 (RAMP1), and calcitonin receptor-like receptor (CLR)]. Injected CGRP causes its activity by engaging these receptors and causing activation of SGCs as demonstrated by an increased expression of GFAP. Pro-inflammatory cytokines IL-1β and IL-6 are released from the SGCs in the TG and these cytokines cause neuronal activation as shown by upregulation of NaV 1.7. Hypothetically, once initiated these effects are self-sustaining because of the formation of a feedback loop due to the secretion of endogenous CGRP and responsible for hyperalgesia as observed at 6 h in the present experiment. Injecting Min (a glial inhibitor) reduced the effect of CGRP, leading to an alleviation of hyperalgesia.

**Figure 10 ijms-20-00711-f010:**
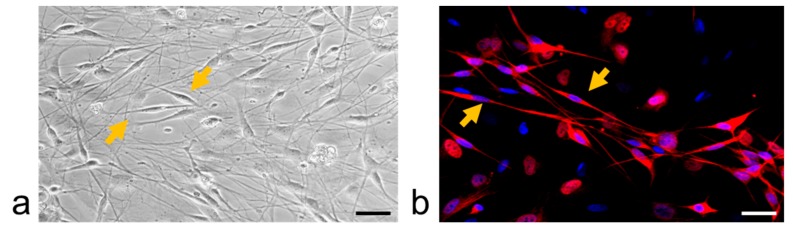
Glial-rich culture, SGCs are identified based on: (**a**) morphology using light microscopy, (**b**) and immunoreactivity as glutamine synthetase (marker of SGC) and DAPI-positive cells using confocal microscopy. Arrow points to the SGCs. Scale bar: 20 µm.

**Table 1 ijms-20-00711-t001:** Average fold change in the level of cytokines release in glial rich cell culture after exposure to CGRP compared to control condition.

Cytokine	Average Fold Change (*n* = 3)	SEM
MIG/CXCL9	6.81	2.84
L-SELECTIN/CD62L/LECAM-1	4.64	2.55
IL-3	4.08	1.25
LIX	3.81	2.12
IL-2	3.10	0.64
IL-6	2.78	0.40
IL-17	2.71	0.90
FRACTALKALINE	2.69	1.40
CNTF	2.63	1.23
MIP-1α/CCL-3	2.51	1.10
IL-1α	2.50	0.75
IL-13	2.38	0.69
IP-10/CXCL10	2.30	0.92
IL-4	2.23	1.37
GM-CSF	1.98	1.51
IL-1ra	1.96	0.31
CINC-2α/β	1.90	0.75
IL-1β	1.86	0.35
IL-10	1.75	0.77
VEGF	1.64	0.46
IFN-ϒ	1.21	0.36
sICAM-1	1.17	0.15
THYMUS CHEMOKINE/CXCL7	1.15	0.69
CINC-3	1.15	0.40
TIMP-1	1.13	0.10
CINC-1	1.13	0.35
RANTES/CCL5	0.95	0.11
TNF-α	0.93	0.45
MIP-3α/CCL20	0.88	0.10

Cytokines released were evaluated in the supernatant using R&D system’s rat cytokine antibody array. 29 cytokines were simultaneously checked for the change in release after stimulation with 1 µM CGRP. The level of 20 cytokines was more than 1.5 fold, which included- IL-1β, IL-6 and IL-1RA, 6 cytokines showed 1–1.5 fold change, and three were below 1 fold. TNF-α expression was found to be below 1 fold change. The average is taken from three independent experiments, and in each experiment TG from three animals were dissociated and passaged to obtain glial rich culture. SEM: Standard error of the mean.

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
