# Peer review of "CGRP Induces Differential Regulation of Cytokines from Satellite Glial Cells in Trigeminal Ganglia and Orofacial Nociception"

_ijms, 2019, doi:10.3390/ijms20030711_

Reviewer 1 Report

Neuro-glia interaction is hot topic in neuroscience and pain. The calcitonin gene-related peptide (CGRP) has been extensively studied for its contribution to nociceptive and chronic pain, notably in migraine. CGRP is usually released by nociceptors into the central nervous system or at the periphery. However, numerous studies have now revealed that neuropeptides, including CGRP, can also be released into sensory ganglia and interact with other cells. CGRP can interact with satellite glial cells (SGCs) that surround nociceptors in sensory ganglia, and it has been shown in vitro that CGRP activate satellite glial cells and stimulate the releases of pro-inflammatory mediators. Here, the authors use intra-ganglionic injections of CGRP to demonstrate that (1) CGRP induces pain and SGC activation, (2) CGRP also increases the expression of the pro-inflammatory cytokine IL-1beta, (3) minocycline (a putative inhibitor of SGCs) reduces CGRP-induced pain, SGC activation, and expression of IL-1beta.

This research is interesting but poorly written. The grammar is ok, but the syntax is dubious and many sentences are too long. The abstract and introduction are confusing. For instance, the expression of CCR2 by SGCs is dubious feature and it is not clear how it support this research. The authors should also better introduce the intra-ganglionic releases of neuropeptides and cytokines in pain, as well as the role of IL-1beta as modulator of Nav1.7 and neuronal excitability.

In the results, please explain what is the L/F ratio and add at least a sentence to explain the rationale of each experiments, especially for the Nav1.7 experiment. It is also surprising the use of minocycline to suppress the activation of SGCs since this drug is very dirty and can inhibit macrophages, MAPKs, and MMP9. It has been shown before that MMP9 can activate SGCs and IL-1beta, so how the authors can exclude an indirect effect of minocycline through this metalloproteinase? Fluorocitrate would have been a better choice as SGC inhibitor. Please add the significance for the cytokine array, and show the original membranes. Add a reference for the use of Tbp as a housekeeping gene.

Discussion should add more comments on the limitations of this study (e.g., minocycline), and the conclusion is pretty weak and should be rewritten to highlights the unique findings of this research.

Author Response

(Please note: Line numbers quoted are with the track changes function on.)

Point 1: Neuro-glia interaction is hot topic in neuroscience and pain. The calcitonin gene-related peptide (CGRP) has been extensively studied for its contribution to nociceptive and chronic pain, notably in migraine. CGRP is usually released by nociceptors into the central nervous system or at the periphery. However, numerous studies have now revealed that neuropeptides, including CGRP, can also be released into sensory ganglia and interact with other cells. CGRP can interact with satellite glial cells (SGCs) that surround nociceptors in sensory ganglia, and it has been shown in vitro that CGRP activate satellite glial cells and stimulate the releases of pro-inflammatory mediators. Here, the authors use intra-ganglionic injections of CGRP to demonstrate that (1) CGRP induces pain and SGC activation, (2) CGRP also increases the expression of the pro-inflammatory cytokine IL-1beta, (3) minocycline (a putative inhibitor of SGCs) reduces CGRP-induced pain, SGC activation, and expression of IL-1beta.

Response 1: We appreciate the reviewer’s comment on our manuscript.

Point 2: This research is interesting but poorly written. The grammar is ok, but the syntax is dubious and many sentences are too long. The abstract and introduction are confusing. For instance, the expression of CCR2 by SGCs is dubious feature and it is not clear how it support this research. The authors should also better introduce the intra-ganglionic releases of neuropeptides and cytokines in pain, as well as the role of IL-1beta as modulator of Nav1.7 and neuronal excitability.

Response 2: The language has been checked and improved. The abstract and introduction have been modified. Intra-ganglionic release of neuropeptides and cytokines and role of cytokines as modulator of NaV1.7 and neuronal excitability have been included in the introduction.

Line no. 22-45. Abstract: “Neuron-glia interactions contribute to pain initiation and sustainment. Intra-ganglionic (IG) secretion of calcitonin gene-related peptide (CGRP) in the trigeminal ganglion (TG) modulates pain transmission through neuron-glia signaling, contributing for various orofacial pain conditions. The present study aimed to investigate the role of satellite glial cells (SGC) in TG in causing cytokine-related orofacial nociception in response to IG administration of CGRP. For that purpose, CGRP alone (10 ml of 10−5 M), Minocycline (5 ml containing 10 mg) followed by CGRP with one hour gap (Min + CGRP) were administered directly inside the TG in independent experiments. Rats were evaluated for thermal hyperalgesia at 6 and 24 hours post-injection using an operant orofacial pain assessment device (OPAD) at three temperatures (37, 45 and 10 °C). Quantitative real time PCR was performed to evaluate the mRNA expression of IL-1β, IL-6, TNF-α, IL-1RA, sodium channel 1.7 (NaV 1.7, for assessment of neuronal activation) and glial fibrillary acidic protein (GFAP, a marker of glial activation). The cytokines released in culture media from purified glial cells were evaluated using antibody cytokine array. IG CGRP caused heat hyperalgesia between 6 - 24 hours (paired-t test, P < 0.05). Between 1 to 6 hours the mRNA and protein expressions of GFAP was increased in parallel with an increase in the mRNA expression of pro-inflammatory cytokines IL-1β and IL-6 and anti-inflammatory cytokine IL-1RA and NaV1.7 (one-way ANOVA followed by Dunnett’s post hoc test, P < 0.05). To investigate whether glial inhibition is useful to prevent nociception symptoms, Minocycline (glial inhibitor) was administered IG 1 hour before CGRP injection. Minocycline reversed CGRP-induced thermal nociception,  glial activity, and down-regulated IL-1β and IL-6 cytokines significantly at 6 hours (t-test, P < 0.05). Purified glial cells in culture showed an increase in release of 20 cytokines after stimulation with CGRP. Our findings support that SGCs in the sensory ganglia contribute for the occurrence of pain via cytokine expression and that glial inhibition can effectively control the development of nociception.”

The following sentences have been removed from the introduction.

Line no. 65-69: “such as activation by monocyte chemotactic protein via the CCR-2 receptor, a unique leukocyte phenotype in the human trigeminal ganglion (TG) and features common to both macrophages and immature myeloid Dendritic Cells. Such characteristics indicate that SGCs exert a role as TG-resident antigen presenting cells with potential T cell modulatory properties”

The following sentences have been added to the introduction.

Line no. 59-62: “Studies, involving in vivo and in vitro settings, have reported that neurotransmitters such as substance P (SP), calcitonin gene-related peptide (CGRP) or adenosine 5'-triphosphate (ATP)) are released within the sensory ganglia due to inflammatory and neuropathic pain (NP) condition.”

Line no. 70-77: “In animal models, results have shown that there is an increased activity of the SGCs and an increase in the cytokine level during pain condition. Release of cytokines from the activated glial cells may be responsible for the persistence of pain by causing neuronal excitation. In an in vitro study, exogenous application of IL-1β to neurons of trigeminal ganglion evoked differential responsiveness from neurons. This effect was shown to be mediated by the modification of voltage gated sodium channels (NaV) and regulated by MAP Kinase thereby contributing to inflammatory hyperalgesia. IL-6 modulates neuronal excitability through NaV1.7, which also involves activation of MAP-kinase pathway.”

Point 3: In the results, please explain what is the L/F ratio and add at least a sentence to explain the rationale of each experiments, especially for the Nav1.7 experiment.

Response 3: The following changes have been made:

Line no. 98-103: The reward-licking events/face-contact events ratio (L/F) and stimulus duration/face-contact events were evaluated after IG drug administration and compared with the baseline. These parameters depend on the contact made by animal with the thermode to reach the reward bottle. As the temperature become aversive or less tolerable, the animal withdraws more frequently, thus, decreasing the L/F ratio and the stimulus duration/stimulus-contact events (seconds).

Line no. 255-261: Studies involving in vivo and in vitro setting have shown that glial secreted cytokines causes neuronal excitation. This neuronal excitability may be due to the modification of NaV channels, out of which NaV1.7 has been reported to be involved. In the present experiment, neuronal excitation was examined by a change in the expression of NaV1.7 after CGRP and Min + CGRP administration.

Point 4: It is also surprising the use of minocycline to suppress the activation of SGCs since this drug is very dirty and can inhibit macrophages, MAPKs, and MMP9. It has been shown before that MMP9 can activate SGCs and IL-1beta, so how the authors can exclude an indirect effect of minocycline through this metalloproteinase? Fluorocitrate would have been a better choice as SGC inhibitor.

Response 4: The inhibitory effect of Minocycline beyond SGC was not ruled out in this study. So an additional discussion about the limitations of present study, which includes Min has been added.

Line no.360-362: Apart from glial inhibition, Minocyline exerts a neuroprotective activity by reducing the expression of metalloproteinases, and an anti-inflammatory activity by inhibiting circulating macrophages. This study did not probe into these possibilities.

Point 5: Please add the significance for the cytokine array, and show the original membranes.

Response 5: The picture of the original membrane has been added with the results (Figure 4). Statistical test was not included because the experiment was performed only three times and the trigeminal ganglion were used from three animals in one experiment, which resulted in the low statistical power of results.

Point 6: Add a reference for the use of Tbp as a housekeeping gene.

Response 6: Reference number 68, line number 454.

Point 7: Discussion should add more comments on the limitations of this study (e.g., minocycline)

Response 7: The limitations of the study have been added.

Line no. 360-368: This study has certain limitations. Apart from glial inhibition, Minocycline has a neuroprotective activity by reducing the expression of metalloproteinases, and an anti-inflammatory activity by inhibiting circulating macrophages. This study did not probe into these possibilities. Therapeutic effect of minocycline was not tested by administering Minocycline after CGRP induced inflammation. Peltier rods of the OPAD were programmed to have a similar bilateral temperature cycle. Therefore, a conclusion about the differential effect of the drug in causing hyperalgesia to the ipsilateral and contralateral side cannot be drawn. However according to one study unilateral injection of CGRP in hind paw had a bilateral effect due to local and neurogenic inflammatory mechanism, and endogenous secretion of CGRP.

Point 8: and the conclusion is pretty weak and should be rewritten to highlights the unique findings of this research.

Response 8: The conclusion has been rewritten.

Line no. 528-537: “In conclusion, we found that IG CGRP injection induces increased sensitivity to heat. This effect was accompanied with an increased SGC and neuronal activation. An increased in expression of cytokine during the same time with in the TG affirm their contribution in pain geneses. Additionally, injecting the glial inhibitor Minocycline 1 hour before CGRP decreased the CGRP induced thermal hyperalgesia, SGC activation and reduced the expression of pro-inflammatory cytokines IL1-β and IL-6 in the TG.  Taken together all these findings support the notion that increased glial activity contributes to hyperalgesia and that glial inhibition can be considered for the management.”

Reviewer 2 Report

Dear Authors, 

The study was well designed and results supported the objectives. There are a few major and minor queries related to the study. 

Major:

1. Why the data presented in Fig. 1 is comparatively more variable at 10- and 37-degree celsius? The data at these temperature showing the trend of analgesia that author did not mention anywhere. Please comment on this part.

2. Discuss the reason behind no change in the status of TNF-alpha in any group in the discussion.

3. How long exactly (hours?) you stimulated the culture cells with CGRP?

4. Schematically represent the mechanism of Min and CGRP interaction using your own and others results. This part is missing from the discussion.

5. Why the authors used the different types of anesthesia instead of only pentobarbital? 

6. The total volume of injection is variable in the different group of animals (5-15 uL). Please explain the reason.

7. The overall study lacks the effect of Minocycline after CGRP related activity. The authors should check the effect of Min. after the CGRP related glial activity or inflammation.   

Minor:

Do not abbreviate the same word, again and again, such as IG, CGRP etc..  

Line 126-write the fig 3 inside the brackets.

Give a table number.

Put the space before any unit.

Correct the long sentences and re-write the immunohistochemistry protocol.

Author Response

(Please note: Line numbers quoted are with the track changes function on.)

Point 1: The study was well designed and results supported the objectives. There are a few major and minor queries related to the study. 

Response1: We appreciate the positive feedback from the reviewer. Our response to each query raised by the reviewer is as follows.

Major:

Point 2: Why the data presented in Fig. 1 is comparatively more variable at 10- and 37-degree celsius? The data at these temperature showing the trend of analgesia that author did not mention anywhere. Please comment on this part.

Response 2: We added it in the result section (line no. 109-110) and discussion (line no. 277-280).

There is an increase in lick/face-contact ratio as well stimulus duration/face-contact events at 37 degree Celsius (24 hours) (figure 1), but this effect did not reach statistical significance. During 24 hours, there is a down-regulation of pro-inflammatory cytokines, which may be responsible for this effect. Based on the findings of present study this trend cannot be confirmed as due to analgesia after 24 hours. In addition, we have mentioned in the discussion, based on the previous studies results.

Line no. 358-359: “Microinjection of CGRP into different areas of the nervous system resulted in differential behavior responsiveness ranging from increased to decreased nociception”. (References 51-57)

However, at 10 degree Celsius L/F ratio is nearly same as baseline and stimulus duration/face-contact events were slightly increased but not statistically significant.

Point 3: Discuss the reason behind no change in the status of TNF-alpha in any group in the discussion.

Response 3:The authors have discussed it between line no. 329-341: “In our experimental condition, TNF-α mRNA and protein expression were relatively unchanged in all the groups. In antibody cytokine array analysis studies, CGRP stimulation of glial rich culture showed different results in the fold change of TNF-α by different researchers- >3-fold increase in TNF-α [8], 50% of control [29], no significant change [28]. This difference in result may be due to the difference in animal species, the age of the animal used for experiment, cell density, the concentration of CGRP or time of stimulation etc. According to a previous study, CGRP mediated immunosuppressive activity is due to suppression of TLR-stimulated dendritic cells TNF-α production by a mechanism involving rapid up-regulation of the transcriptional repressor inducible cAMP early repressor [41]. In contrast, in an organ culture study of TG, TNF-α mRNA showed highly significant upregulation when co-incubated with CGRP, which was counteracted by the addition of CGRP8-37 (CGRP antagonist) [42]. TNF-α induces pro-inflammatory signal cascades accompanied with an increase in the synthesis and release of CGRP by trigeminal ganglion neurons [43] and in migraineurs TNF- α and IL-1β are shown to be elevated during the attack [44]. However, the reason behind no change in status of TNF-α in our study requires more research and can be the direction of our future study.

Point 4: How long exactly (hours?) you stimulated the culture cells with CGRP?

Response 4: Twelve hours- overnight (added to method section, Line no. 509).

Point 5: Schematically represent the mechanism of Min and CGRP interaction using your own and others results. This part is missing from the discussion.

Response 5: Figure 9 added in the discussion.

Point 6: Why the authors used the different types of anesthesia instead of only pentobarbital? 

Response 6: All the animals were euthanized by using an overdose of pentobarbital. However, for behavior testing and other assays agonist-antagonist drugs were utilized to effect early recovery of animals from anaesthesia after intraganglionar injections.

Point 7: The total volume of injection is variable in the different group of animals (5-15 uL). Please explain the reason.

Response 7: The saline and CGRP injected group received 10 µL volume. Whereas Minocycline + CGRP injected group received 5 µL followed by 10 µL volumes of drug. A gap of one hour was maintained between Min and CGRP injection to prevent the effect of volume injection. So at one time maximum injected volume was only 10 µL for each group.

Point 8: The overall study lacks the effect of Minocycline after CGRP related activity. The authors should check the effect of Min. after the CGRP related glial activity or inflammation.   

Response 8: The primary aim of the study was to check the effect of glial activation on the thermal hyperalgesia due to cytokines produced inside the TG. In addition, to support the hypotheses that origin of these cytokines is SGCs, glial inhibition was undertaken before CGRP injection, as seen in some other previous reports (Gong et al. 2015; Liu et al. 2010, references quoted below for the reviewer’s perusal). Therapeutic efficacy of Minocycline in controlling CGRP induced inflammatory changes were not a part of study design that was the reason this was not tested in the present study. However, understanding the importance of this step, we added this as a limitation of the study in discussion section (Line no. 363).

1.     Kerui Gong, Xiaoju Zou, Perry N Fuchs and Qing Lin. Minocycline inhibits neurogenic inflammation by blocking the effects of tumor necrosis factor-α. Clinical and Experimental Pharmacology and Physiology (2015) 42, 940–949

2.     Cui-Cui Liu, Ning Lu, Yu Cui, Tao Yang, Zhi-Qi Zhao, Wen-Jun Xin, Xian-Guo Liu. Prevention of Paclitaxel-induced allodynia by Minocycline: Effect on loss of peripheral nerve fibers and infiltration of macrophages in rats. Molecular Pain 2010, 6:76

Minor:

Point 9: Do not abbreviate the same word, again and again, such as IG, CGRP etc.

Response 9: Changes have been made.

Point 10: Line 126-write the fig 3 inside the brackets.

Response 10: Changes have been made.

Point 11: Give a table number.

Response 11: Table number has been assigned.

Point 12: Put the space before any unit.

Response 12: Changes have been made.

Point 13: Correct the long sentences and re-write the immunohistochemistry protocol.

Response 13: Changes have been made and immnohistochemistry protocol has been rewritten.

Reviewer 3 Report

This is a very interesting study that evaluates the neuron-glia cross-talk within the trigeminal ganglion by using an original approach, i.e. the intraganglionic injection of CGRP, and demonstrates the generation of pain which is linked to the activation of satellite glial cells by CGRP with the consequent release of a mixture of cytokines.

While the communication between sensory neurons and glial cells within the trigeminal ganglion has been the object of several previous studies, this unusual approach adds some more hints to the whole scenario; additionally, the injection of minocycline provides a direct confirmation of the role of glial cells in pain.

I have some questions and comments that need to be addressed to further increase the quality of the paper.

1.     Figure 1: at variance from data shown in the subsequent figures the behavioral outcome at 1 hour has not been evaluated, why? Additionally, what is the possible explanation for the trend to increase in the two parameters at 24 hours? Are animals less stressed for the injection than at 6 hours? Please, add a comment on this issue.

2.     Since the intraganglionic injection of CGRP is not a “classical” pain model, how was the concentration chosen? What is the actual concentration which is reached within the ganglion? Is there a correlation with the CGRP concentrations observed under some pathological conditions? How were the time points chosen? Is there a reference available to be quoted on this Method?

3.     Figure 3 and Table (please, the only one table must be numbered as well): authors have correctly compared the release of cytokines at various time points after CGRP injection with saline-injected animals. What about a comparison between ipsi- and contra-lateral ganglia in CGRP-injected animals? These data would further confirm the conclusions raised by the authors.

4.     Culturing trigeminal cells from adult animals is quite unusual and not easy to be done. Please, add a representative image of both mixed neuron-glia and purified glia cultures.

5.     Some English corrections must be made; for example, line 146 “controlled” is incorrect; the first sentence in line 173 is quite confused.

6.     I have some uncertainties on data shown in Figure 7. Although they are interesting, they seem quite disconnected from the rest of the manuscript. What is their aim? Why were these receptors chosen? What about TRPV1 expression? Adding immunohistochemical data showing, if possible, increased Nav1.7 receptor expression in satellite glial cells-encircled neurons would further increase the value of the data provided in this manuscript.

7.     Lines 305-306: the last sentence of the Discussion is useless.

Author Response

(Please note: Line numbers quoted are with the track changes function on.)

Point 1: This is a very interesting study that evaluates the neuron-glia cross-talk within the trigeminal ganglion by using an original approach, i.e. the intraganglionic injection of CGRP, and demonstrates the generation of pain which is linked to the activation of satellite glial cells by CGRP with the consequent release of a mixture of cytokines.

While the communication between sensory neurons and glial cells within the trigeminal ganglion has been the object of several previous studies, this unusual approach adds some more hints to the whole scenario; additionally, the injection of minocycline provides a direct confirmation of the role of glial cells in pain.

I have some questions and comments that need to be addressed to further increase the quality of the paper.

Response 1: Thank you for reviewing the manuscript; we appreciate a positive feedback from the reviewer. Our response to each query raised by the reviewer are as follows.

Point 2: Figure 1: at variance from data shown in the subsequent figures the behavioral outcome at 1 hour has not been evaluated, why? Additionally, what is the possible explanation for the trend to increase in the two parameters at 24 hours? Are animals less stressed for the injection than at 6 hours? Please, add a comment on this issue.

Response 2: One-hour post injection animals exhibited less activity in all the groups and it was affecting the animals’ voluntary feeding activity. This may be attributed to incomplete recovery from anesthesia, and was the primary reason that this time  was not selected for behavior assessment.

There is an increase in lick/face-contact ratio as well stimulus duration/face-contact events at 37 degree Celsius (24 hours) (figure 1), but this effect did not reach statistical significance. During 24 hours, there is a down-regulation of pro-inflammatory cytokines, which may be responsible for this effect. Based on the findings of present study this trend cannot be confirmed as due to analgesia after 24 hours. In addition, we have mentioned in the discussion, based on the previous studies results-

Line no. 358-359: “Microinjection of CGRP into different areas of the nervous system resulted in differential behavior responsiveness ranging from increased to decreased nociception”. (References 51-57)

Point 3: Since the intraganglionic injection of CGRP is not a “classical” pain model, how was the concentration chosen? What is the actual concentration which is reached within the ganglion? Is there a correlation with the CGRP concentrations observed under some pathological conditions? How were the time s chosen? Is there a reference available to be quoted on this Method?

Response 3: Concentration of the drug is chosen from a previously published research (reference no. 34, added to the method section). In the present study, we did not evaluate the actual concentration of the drug, which reached the ganglion. There is some data from human and animal studies indicating that the amount of CGRP in the tissues and exudate in normal and pathological condition may range from nano molar to micro molar concentrations (Alstergren et al. 1995; Petersson et al. 1989). Following references are quoted here for reviewer’s perusal:

1.     Alstergren P, Appelgren A, Appelgren B, Kopp S, Lundeberg T, Theodorsson E. Co-variation of neuropeptide Y, calcitonin gene-related peptide, substance P and neurokinin A in joint fluid from patients with temporomandibular joint arthritis. Arch Oral Biol. 1995 Feb;40(2):127-35.

2.     Petersson G, Malm L, Ekman R, Håkanson R. Capsaicin evokes secretion of nasal fluid and depletes substance P and calcitonin gene‐related peptide from the nasal mucosa in the rat. British Journal of Pharmacology, 1989, 98: 930-936. doi:10.1111/j.1476-5381.1989.tb14623.x

Times were decided on the basis of present experimental condition and based on the work of other author who also used intraganglionar drug administration and behavior testing (reference no. 23 added to the method)

Point 4: Figure 3 and Table (please, the only one table must be numbered as well): authors have correctly compared the release of cytokines at various time s after CGRP injection with saline-injected animals. What about a comparison between ipsi- and contra-lateral ganglia in CGRP-injected animals? These data would further confirm the conclusions raised by the authors.

Response 4: Table number has been assigned (Table 1). We have added the comparison between ipsi and contralateral data to Figure 3 (b).

Point 5: Culturing trigeminal cells from adult animals is quite unusual and not easy to be done. Please, add a representative image of both mixed neuron-glia and purified glia cultures.

Response 5: Images of purified glial cultures using light microscopy as well as confocal microscopy showing presence of satellite glial cells as Glutamine synthetase positive cells have been added (Figure 10). However, the neuron-glia mixed culture pictures are not high quality (for publication), so they were not added. It is pasted here for reviewer’s perusal-

Point 5: Some English corrections must be made; for example, line 146 “controlled” is incorrect; the first sentence in line 173 is quite confused.

Response 5: Corrections have been made; sentence have been revised and modified.

Point 6: I have some uncertainties on data shown in Figure 7. Although they are interesting, they seem quite disconnected from the rest of the manuscript. What is their aim? Why were these receptors chosen? What about TRPV1 expression? Adding immunohistochemical data showing, if possible, increased Nav1.7 receptor expression in satellite glial cells-encircled neurons would further increase the value of the data provided in this manuscript.

Response 6: We have clarified the rationale of using NaV 1.7 expression changes by incorporating more information in the introduction, results and discussion.

Introduction line no.72-77: “The release of cytokines from the activated glial cells may be responsible for the persistence of pain by causing neuronal excitation. In an in vitro study, exogenous application of IL-1β to neurons of trigeminal ganglion evoked differential responsiveness from neurons. This effect was shown to be mediated by the modification of voltage-gated sodium channels (NaV) and regulated by the MAPK thereby contributing to inflammatory hyperalgesia [11,12]. IL-6 modulates neuronal excitability through NaV1.7, which also involves activation of the MAPK pathway.”

Results line no. 257-261: “Studies involving in vivo and in vitro setting have shown that glial secreted cytokines cause neuronal excitation. This neuronal excitability may be due to the modification of NaV channels, out of which NaV1.7 has been reported to be involved [11-13]. In the present experiment, neuronal excitation was examined by a change in the expression of NaV1.7 after CGRP and Min + CGRP administration.”

We tried the staining for neuronal activation (Activating transcription factor 3 (ATF 3)). However, in our hands ATF3 was found to be localized in neuronal cytosol and not in neuronal nuclei as shown by other researchers. The reason for it may be non-specific staining, and that is why we did not include it in our results.

Point 7: Lines 305-306: the last sentence of the Discussion is useless.

Response 7: The sentences have been deleted.

Round  2

Reviewer 1 Report

The authors have successfully addressed my major concerns.

Reviewer 2 Report

None

Reviewer 3 Report

I am fully satisfied with authors’ answers to my comments and suggestions